# BALANCING LABEL QUANTITY AND QUALITY FOR SCALABLE ELICITATION

## ABSTRACT

Scalable oversight studies methods of training and evaluating AI systems in domains where human judgement is unreliable or expensive, such as scientific research and software engineering in complex codebases. Recent work in this area by Burns et al. (2023) suggests that Language Models (LMs) pretrained on internet-scale corpora exhibit an inductive bias toward producing correct answers, even when finetuned on error-prone labels produced by a smaller language model. This suggests that massive pretraining combined with finetuning on imperfect human labels may be a solid baseline method for scalable oversight. In the real world, however, label quality is not fixed: practitioners face a *quantity-quality tradeoff* when generating finetuning data. In this paper, we explore the microeconomics of the quantity-quality tradeoff on binary NLP classification tasks used in Burns et al. (2023). We find that there are three regimes of eliciting classification knowledge from pretrained models using supervised finetuning: quantity-dominant, quality-dominant, and a mixed regime involving the use of low- and high-quality data together to attain higher accuracy at a lower cost than using either alone. We explore sample-efficient elicitation methods that make use of two datasets of differing qualities, and establish a Pareto frontier of scalable elicitation methods that optimally trade off labeling cost and classifier performance.

## 1 INTRODUCTION

While supervised learning and reinforcement learning from human feedback (Stiennon et al., 2022) have been effective techniques for training LMs, recent models and benchmarks have required increasing investments in subject-matter experts for annotation and red-teaming (OpenAI, 2023; Rein et al., 2023). Scalable oversight studies methods of training and evaluating AI systems in domains where accurate feedback is limited because of cost.

The definition of scalable oversight we use in this paper mirrors the original definition from Amodei et al. (2016)[1], which describes scalable oversight as a quantitative problem aimed at reducing the cost of high quality supervision (Shlegeris, 2024). We find this framing useful for thinking about supervising AI systems with advanced capabilities, such as automating the core activities of AI research: How can you reduce the cost of eliciting a capability from a model?

For example, when supervising a system to write complex software, you might like to elicit the model's knowledge of whether there are security vulnerabilities in the code. It would be extremely expensive to attain high-quality labels of secure and subtly-insecure code, especially if the AI-written software is significantly out-of-distribution relative to prior known vulnerabilities. This means it would be crucial to know how sample-efficient learning will be, and to strike the right balance between label quality and quantity.

Amodei et al. (2016) discusses these issues in the context of a reinforcement learning (RL) agent "given limited access to the true objective function," proposing many promising and since-proven directions including reward modeling, active learning (explored here), and unsupervised learning (cf.

---

[1]While some, including Burns et al., consider weak-to-strong generalization a complement to scalable oversight (Radhakrishnan et al., 2023) rather than a scalable oversight approach *per se*, the pragmatic definition we adapt from Amodei et al. (2016) encompasses weak-to-strong generalization and the methods introduced in this paper.

the role of pretraining in weak-to-strong generalization). We focus on the binary classification setting because it is simple and informative for many practical cases of evaluating complex AI actions.

Burns et al. (2023) studies finetuning methods that make use of unreliable labels (often less than 90% accurate on their binary classification datasets). Their finding of "weak-to-strong generalization," in which finetuning on low-accuracy "weak" labels can elicit higher accuracy classifications from strong pretrained models, is a prominent research direction for scalably supervising models. However, Burns et al. (2023) does not explore strategies that allocate some budget to fewer, higher-quality labels, which, as we show, are more effective for a variety of realistic economic settings.

Our contributions are as follows:

1. We demonstrate that there exists an important elicitation regime that substantially benefits from using a combination of low-quality and high-quality labels, rather than either alone.

2. We empirically and quantitatively characterize the quantity-quality tradeoff for a range of datasets, microeconomic assumptions, and model scales.

3. We propose the research framing of reducing the cost of eliciting knowledge from capable models, and establish a Pareto frontier of scalable elicitation methods that maximize classification accuracy and minimize labeling cost.

Our work aims to be agnostic to the details of the scalable oversight problem, so we experiment with a variety of datasets and assumptions about labeling costs.

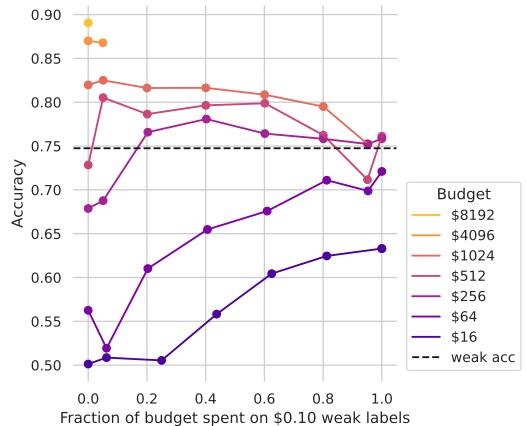

Figure 1: Illustration of the tradeoff between quantity and quality of labels for sequential SFT. We arbitrarily define the cost of a high-quality label to be \$1. Points lying on the y-axis can be understood as the accuracy attained when finetuning exclusively on high-quality labels as usual, for each budget. Along the x-axis, one high-quality label is given up for every 10 weak labels used because they cost \$0.10. Weak labels are generated by Qwen1.5 0.5B (Bai et al., 2023), and the strong model, Llama 3 8B (Dubey et al., 2024), is sequentially trained on weak then high-quality labels. **When the budget is not large enough to attain >0.8 accuracy using high-quality labels alone, accuracy can be improved by spending some or all budget on a large quantity of weak labels.** Results are averaged over 5 binary classification tasks (Hellaswag, SciQ, CosmosQA, Quail, and SocialIQA). Missing points from the top few lines are due to some datasets not having enough available examples. Note that the weak label accuracy is measured on the train set, which is not necessarily distributed identically to test.

## 2 THREE REGIMES OF ELICITATION

We find that there are three regimes of eliciting classification knowledge using supervised finetuning (SFT), depending on how many labels are affordable.

**Quality-dominant**. You can afford many high-quality examples—enough to train to near convergence—and your best strategy is to invest only in these. This is the bread-and-butter of present-day ML practitioners.

**Quantity-dominant**. You cannot afford almost any high-quality examples, but neither can you afford enough weak examples to train to near convergence, so every marginal dollar[2] is best spent on weak labels.

**Mixed**. You cannot afford a large enough quantity of high-quality examples to train to near convergence, but you can afford enough weak examples. We find that at first, because the weak labels have non-trivial accuracy (and to some extent because of weak-to-strong generalization), weak labels update the model in the desired direction. Then, after

---

[2]Our convention in this paper will be to use a fictitious currency, denoted \$, that is tied to the cost of labeling one high-quality example. In reality we are targeting problems where each label costs orders of magnitude more than 1 USD.

training on enough weak examples to approach convergence, the marginal benefit of a dollar spent on weak labels decreases below the marginal benefit of spending on high-quality labels. In this regime, it is optimal to spend some budget on a large volume of low-quality labels and some budget on high-quality labels.

This paper focuses on the mixed regime, in which the optimal allocation of labeling resources is not *a priori* evident. We begin by empirically demonstrating the three regimes in a simple training strategy we call sequential SFT (Sec. 3.2). Then we consider a wide range of sample-efficient elicitation methods to make prescriptions about the optimal method and quantity-quality tradeoff in various circumstances.

# 3 METHODS

## 3.1 DATA

We experiment on a variety of binarized NLP classification tasks, largely mirroring a subset of the tasks used in Burns et al. (2023). We look at BoolQ, HellaSwag, SciQ, Cola, CosmosQA, QuAIL, and SocialIQA.

Like Burns et al. (2023), we generate weak labels using small LMs that have been finetuned on the task. Specifically, we train the weak model on 8,000 ground-truth-labeled examples for 3 epochs, and gather the weak model's probabilities on those 8,000 examples along with 50,500 new examples to form the train/val pool (or however many are available after making the test split). This pool is balanced, but the training and validation sets sampled from it are not necessarily balanced.

Models are tested on a balanced, held-out test set. Note that not all datasets we use have i.i.d. train and test splits. The covariate shift between train and test is relatively minor (we are using standard NLP tasks), but means that weak label accuracy cannot be perfectly interpreted as the accuracy on the target task.

## 3.2 ELICITATION METHODS

We only consider methods that make use of one or two data sources for simplicity.

**Sequential SFT** first trains the strong model on weak labels using supervised finetuning (SFT) with LoRA, then finetunes on a disjoint set of high-quality examples. Both finetuning stages early-stop based on validation AUROC. The train and validation sets for each stage are i.i.d., and both are counted toward the labeling budget. When zero weak examples or zero high-quality examples are used, the corresponding stage is

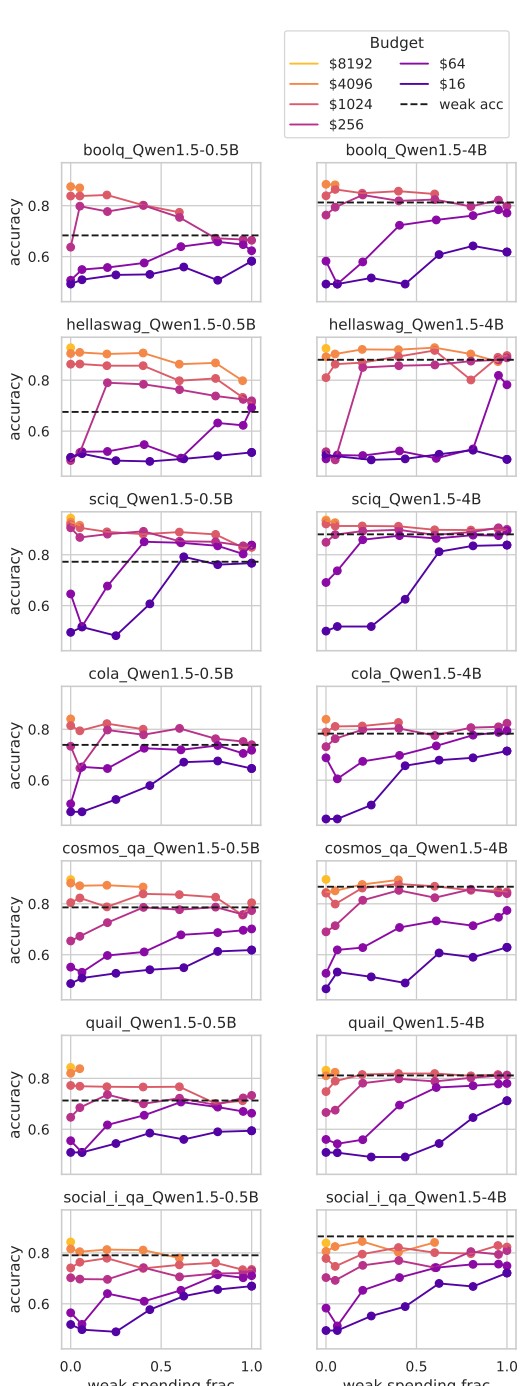

Figure 2: Comparison between weak labels generated by Qwen1.5 0.5B vs Qwen1.5 4B at a weak marginal cost of $0.10.

skipped. We randomly initialize a new head for training. For additional training details, see Appendix A.

**Few-shot prompting.** This method utilizes LMs' in-context learning abilities (Brown et al., 2020). The few-shot examples in the context are shuffled at each inference, and use "0" and "1" as class label tokens.

**Few-shot-prompted sequential SFT.** This method uses sequential SFT on a distribution of few-shot prompts with the aim of increasing the sample-efficiency of SFT by increasing the task's salience. In Figure 4, we experiment with varying the quantity of in-context examples, and whether the in-context examples and SFT examples are weak or high-quality. We observe that the kind and quantity of in-context examples is relatively inconsequential, so we primarily experiment with **2-shot-prompted sequential SFT**, where the in-context examples are both weak.

**Uncertainty sampling.** Inspired by the active-learning literature Kolossov et al. (2023); Gal et al. (2017), we experiment with a variant of sequential SFT that samples high-quality data for labeling in the second stage based on the confidence of the model after the first stage of (weak) training. Specifically, we deterministically select the examples where the model's prediction entropy is highest (i.e., where the probability it assigns to the positive class is closest to 0.5) at the beginning of the second stage. This method has the important practical limitation that it requires labeling in between the two stages of training, which can subsantially slow down the finetuning process, and that it may pose additional costs to search for examples where the model is uncertain.

**Log-confidence auxiliary loss.** Burns et al. (2023) found that a certain confidence auxiliary loss improves weak-to-strong generalization performance. We experiment with a version of sequential SFT that uses this loss function (with a minibatch size[3] of 8) during the weak stage of training.

Note that some methods have inherent limitations in what dataset sizes they can be used with. For example, sequential SFT is not well-equipped for datasets with less than a dozen examples distributed across the train and validation sets, while few-shot in-context learning is, but suffers memory and context-length issues for large datasets.

We aim to test elicitation methods that are general: they can be used for arbitrary classification tasks of which the subject model has implicit knowledge, regardless of how similar that knowledge looks to common natural language tasks. Unfortunately, most capabilities tested in current NLP benchmarks are well-represented in natural language pre-training, marking a limitation of studying the generalizability of some methods, especially prompting-based methods.

## 4 RESULTS

Figure 1 is a demonstration of the quantity-quality tradeoff for sequential SFT for the setting where weak labels (from Qwen1.5 0.5B) are assumed to be 10x cheaper than high-quality labels. We see the "quantity-dominant" regime for budgets of $\leq$\$64 (not enough labels can be afforded to approach convergence even when all budget is spent on weak labels), the "mixed" regime for budgets \$256-\$512 (there are enough weak examples to converge, but not enough high-quality labels), and the "quality-dominant" regime for budgets of at least \$1024 (it is optimal to use only high-quality labels). In the "mixed" regime the optimal budget allocation involves a large quantity of weak labels, as well as some high-quality labels.

Figure 2 breaks down the sequential SFT results by dataset, and varies the quality of weak labels. Because the qualitative results are not very sensitive to weak label cost (see Figure 5), we focus on \$0.10 weak labels for readability. We find, as expected, that higher-quality weak labels are useful in a wider range of circumstances (though the effect-size is small) and that weak labels are useful for a variety of datasets and weak label qualities.

### 4.1 SCALING

Do the three regimes persist with scaling? We experiment with sequential SFT on MMLU Hendrycks et al. (2021) using Llama-3 8B base, Llama-3 70B base, and GPT-4o-mini-2024-07-18. The OpenAI

---

[3]Because the log-confidence loss is minibatch-dependent, this is an important hyperparameter. We set it to the largest size within VRAM constraints.

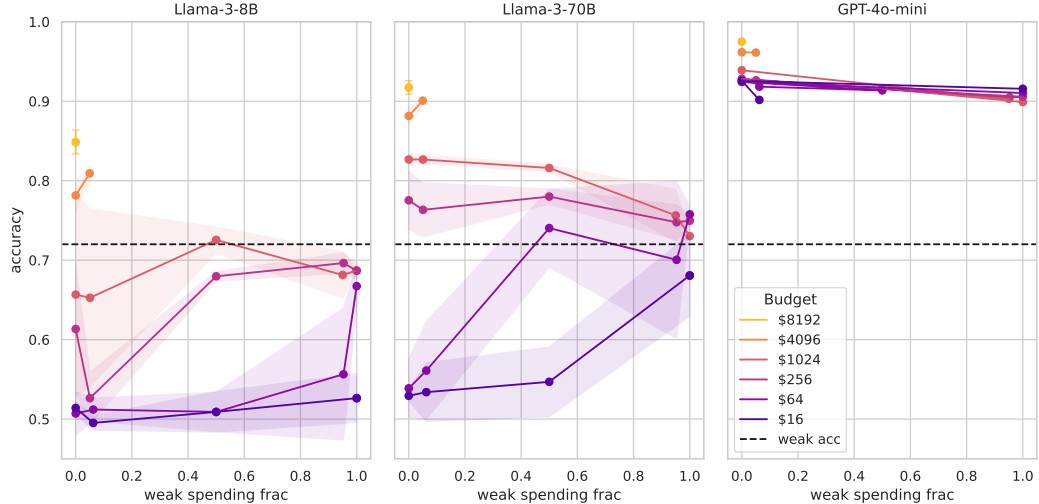

Figure 3: Scaling trends of sequential SFT on MMLU (without early-stopping as described in Sec 4.1). Weak labels are 70.2% accurate and generated by davinci-002, which is less capable than Llama-3-8B. Weak labels are again assumed to cost 10 times less than high-quality labels. Errorbars are standard deviations over random seeds. We use 3 random seeds, except for training runs where the smaller stage takes less than or equal to 10 examples, in which case we use 7 random seeds. We see weak evidence corroborating prior work that suggests larger models require fewer finetuning examples to elicit their knowledge (Zhang et al., 2024). High accuracy in MMLU can be elicited from GPT-4o-mini even with 16 finetuning examples.

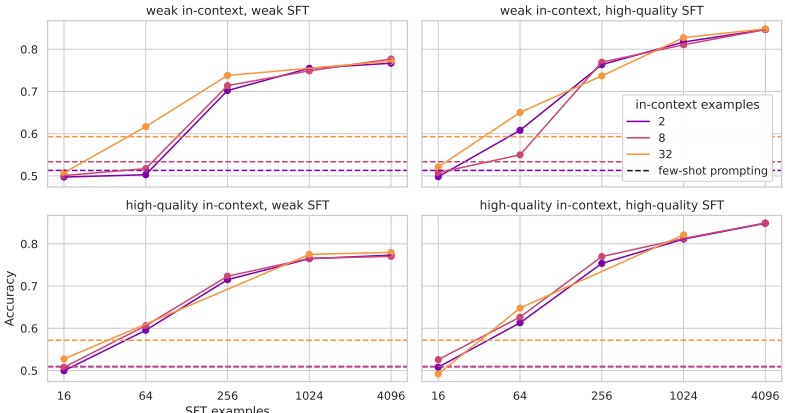

Figure 4: Few-shot-prompted SFT with various quantities of weak and high-quality labels in-context and used for SFT. The quality of in-context examples is inconsequential, while the quality of SFT examples matters substantially.

finetuning API does not allow for early-stopping, so in an effort to make the experiment as controlled as is possible with commercial models, we modify the sequential SFT training setup for Llama to more closely mirror OpenAI's. This primarily involves training with a batch size and number of epochs determined based on the number of training examples, as described in Appendix A. We are also unable to randomly initialize a new head, so for GPT-4o-mini only, we use the difference between the "Yes" and "No" logits.

Figure 3 shows how the quantity-quality tradeoff changes as model scale increases for a fixed task using sequential SFT. Larger models are more sample efficient which correspondingly reduces the cost of elicitation. 256 and 1024 high-quality finetuning examples do not reliably elicit knowledge from Llama-3-8B, but elicit most of Llama-3-70B's knowledge. We were not able to find a quantity of

Table 1: Percent accuracy (optimal weak label fraction). Tabular form of Figure 5 at $0.10 weak labels. Errorbars are standard deviations over 3 random seeds, macro-averaged over datasets. Each accuracy is the highest average accuracy (over datasets and seeds) that can be attained with a cost less than or equal to the budget, with parentheses showing the fraction of labels that should be low-quality to optimize performance.

| Budget | $5 | $17 | $65 | $257 | $1025 | $4097 |
|---|---|---|---|---|---|---|
| Seq SFT | - | 60±3 (1.0) | 70±2 (1.0) | 77±2 (0.9) | 82±2 (0.3) | 87±1 (0.0) |
| +2-shot ICL | - | **63±7 (1.0)** | **75±2 (1.0)** | 77±3 (0.9) | **84±1 (0.3)** | **88±1 (0.0)** |
| +log-conf. | - | 59±2 (1.0) | 69±3 (1.0) | 76±3 (0.9) | 82±2 (0.9) | 86±1 (0.0) |
| +unc. sampl. | - | 60±2 (1.0) | 70±2 (1.0) | **79±1 (0.9)** | 82±2 (0.9) | 87±1 (0.0) |
| few-shot ICL | **58±5 (1.0)** | 58±5 (1.0) | 58±5 (1.0) | 58±5 (1.0) | 58±5 (1.0) | 58±5 (1.0) |

high-quality finetuning examples that cause GPT-4o-mini to leave the "quality-dominant" elicitation regime because the OpenAI finetuning API requires at least 10 examples, which is enough for 0.92 accuracy. This may be due GPT-4o-mini's large scale, or confounders such as optimizations in OpenAI's finetuning service, using the existing LM head rather than a new head, or post-training enhancements that make MMLU especially easy to elicit. The scaling results for sequential SFT suggest that for a fixed labeling budget and task, the quantity-quality tradeoff weighs more in favor of quantity the smaller the model. Our results are weak evidence that the "mixed" regime exists across model scales at decreasing budgets, even though we were not able to test this hypothesis for GPT-4o-mini.

## 4.2 COMPARISON OF METHODS

We turn our attention toward finding the optimal elicitation method (listed in Sec. 3.2) for various budgets and weak label costs.

First, Figure 4 compares ways of making use of the weak and high-quality labels in few-shot-prompted SFT. The quality (and to some extent quantity) of the few-shot examples turn out to be relatively inconsequential, in line with Min et al. (2022), while high-quality labels are important for finetuning. For this reason our main few-shot-prompted SFT experiments in Figure 5 use 2-shot prompts with weak labels.

The optimal methods can be seen in Figure 5, which shows the Pareto frontier of finetuning strategies for three different hypothetical weak label costs. Results broken down by each of the three datasets can be found in Appendix figures 6, 7, and 8, suggesting that the results hold across tasks and weak label qualities. Results for all methods including ones not on the Pareto frontier (sequential SFT and log-confidence) can be seen in Table 1. We find that log-confidence loss is not particularly effective, which is in line with results from the smaller models used in Burns et al. (2023) and a follow-up by Scherlis et al. (2024). Uncertainty sampling the high-quality labels can be effective when the budget is just large enough that you should train with more than just weak labels — that is, the low-budget end of the "mixed" regime.

Overall, methods making use of LMs' in-context learning abilities are most effective, with standard few-shot prompting being optimal in extremely data-poor regimes, and 2-shot-prompted sequential SFT being optimal when a larger quantity of labels are available.

## 5 RELATED WORK

**Scalable oversight.** There exists a variety of work in scalable oversight that aims to **amplify** human labelers with AI assistants to improve supervision quality Saunders et al. (2022). Because it is impractical to evaluate scalable oversight techniques in domains where humans don't provide reliable answers, the **sandwiching** paradigm was proposed in Cotra (2021) and developed in Bowman et al. (2022), in which non-expert or artificially hindered human annotators are tasked with supervising a capable model. In AI **debate** (Irving et al., 2018; Michael et al., 2023), two capable but untrusted AI systems compete to persuade a human judge. Recent experiments have found that debates between more persuasive AI debaters result in higher quality judgements by an artificially hindered

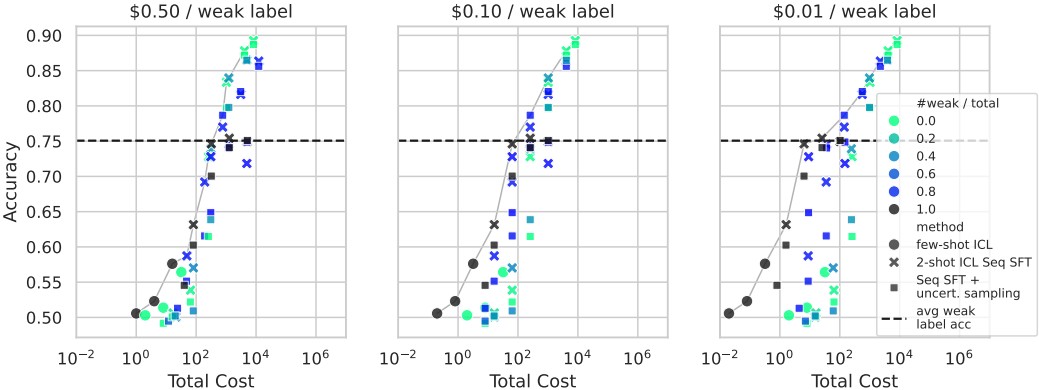

Figure 5: Accuracy vs cost of the top three finetuning methods, at three different weak label costs, with weak labels generated by Qwen1.5 0.5B. Each point is the average accuracy over Hellaswag, SocialIQA, and CosmosQA. The color indicates the fraction of labels that are weak, with black indicating that exactly zero high-quality labels were used. The Pareto frontier is shown in gray. 2-shot-prompted sequential SFT makes sample-efficient use of labels, making it the most effective method for most budgets. For low budgets, however, few-shot prompting with weak labels is most effective.

judge (Khan et al., 2024). Our work, on the other hand, focuses on making most effective use of limited supervision to maximally elicit model capabilities, which is more directly related to empirical Eliciting Latent Knowledge (Christiano et al., 2021) works such as Burns et al. (2022; 2023); Roger et al. (2023) and Mallen et al. (2024). These papers distinguish themselves from the aforementioned scalable oversight directions in their focus on the empirical generalization properties of training with limited supervision.

**Few-shot learning.** Few-shot learning aims to make effective use of a small amount of labeled data. Large LMs are well-known to possess impressive few-shot in-context learning abilities (Brown et al., 2020; Min et al., 2022). Some existing few-shot learning methods make use of auxiliary, off-task, data to improve LM few-shot learning performance (Albalak et al., 2024; Aghajanyan et al., 2021; Esfandiarpoor et al., 2020). These auxiliary data sources can be understood as somewhat analogous to the weak datasets used in this work. For a thorough overview of the few-shot learning literature, not limited to LMs, see Parnami & Lee (2022).

**Data selection.** Several existing works aim to make decisions about how much of various data sources to use (Albalak et al., 2023; Xie et al., 2023; Siddiqui et al., 2022; Sorscher et al., 2022; Abbas et al., 2023). These typically focus on pre-training rather than finetuning, and make data selection decisions under a *computing cost* constraint rather than a *labeling cost* constraint.

## 6 DISCUSSION AND FUTURE WORK

In this paper we empirically characterized the quantity-quality tradeoff for a variety of datasets and microeconomic assumptions, and then established a Pareto frontier of inexpensive and performant elicitation methods. As continued research expands and strengthens this Pareto frontier, our ability to reliably supervise complex actions from advanced AI systems improves.

We focus this paper on "elicitation," but it can be unclear when SFT is best understood as eliciting a capability that was "already there," as opposed to learning a new capability. However, we argue that the tasks considered in this paper — and many real-world tasks — are best understood as elicitation. We often observe in this paper that finetuning a model on a few dozen or hundred question-answer pairs causes the model to answer new, semantically unrelated, questions with nontrivial accuracy. The weights learned during pretraining already approximately encode the function that maps questions to correct answers, and finetuning causes the model to transmit this knowledge in its output.

Our work is limited to binary classification tasks. Although binary classification subsumes a wide variety of practical use-cases, we expect there may be additional challenges with eliciting knowledge in settings with wide output spaces (e.g. generative or reinforcement learning tasks) such as exploration and sparse reward. More generally, it is unclear how analogous our settings are to practical settings that challenge human experts.

One notable limitation is that we do not compare finetuning methods aimed at eliciting highly reliable knowledge (i.e., >99% accurate) because we do not use reliable enough benchmarks to measure very high accuracy. High-quality labels might be more important in this regime to clarify edge cases, or less important because the model has a salient and well-generalizing representation of the task that is easy to elicit.

Our paper is broadly aimed at expanding the Pareto frontier of elicitation accuracy and cost. To this end, we explored a variety of finetuning methods that make use of a combination of high-quality labels and inexpensive weak labels. However, there are many other avenues that can be explored to expand this Pareto frontier, such as easy-to-hard and domain generalization.

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

Table 2: Hyperparameters used in scaling experiments to mimic OpenAI finetuning API

| dataset size ($n$) | batch size | number of epochs |
|---|---|---|
| $n < 30$ | 1 | $\lceil 100/n \rceil$ |
| $30 \leq n < 1,024$ | 1 | 3 |
| $1,024 \leq n < 4,096$ | 2 | 3 |
| $4,096 \leq n < 16,384$ | 8 | 2 |
| $n \geq 16,384$ | 8 | 1 |

Sang Michael Xie, Hieu Pham, Xuanyi Dong, Nan Du, Hanxiao Liu, Yifeng Lu, Percy S Liang, Quoc V Le, Tengyu Ma, and Adams Wei Yu. Doremi: Optimizing data mixtures speeds up language model pretraining. In A. Oh, T. Naumann, A. Globerson, K. Saenko, M. Hardt, and S. Levine (eds.), *Advances in Neural Information Processing Systems*, volume 36, pp. 69798–69818. Curran Associates, Inc., 2023. URL https://proceedings.neurips.cc/paper_files/paper/2023/file/dcba6be91359358c2355cd920da3fcbd-Paper-Conference.pdf.

Biao Zhang, Zhongtao Liu, Colin Cherry, and Orhan Firat. When scaling meets llm finetuning: The effect of data, model and finetuning method, 2024. URL https://arxiv.org/abs/2402.17193.

All datasets?

# A  METHODS

## A.1  SEQUENTIAL SFT TRAINING DETAILS

The Adam buffer is re-estimated at each training stage, with a linear warmup of 40 steps (**?**), or the number of steps per epoch if that is smaller (because subsequent epochs do not improve the estimate).

When performing early-stopping, we evaluate and save the model every epoch or every 50 steps, whichever is more frequent. Training is terminated after 4 consecutive evaluations that fail to improve upon the best-yet validation AUROC by at least 0.01, and then the checkpoint with the highest validation AUROC is loaded.

We use a cosine learning rate schedule with 625 steps of training per stage (modulo early stopping), except for in our scaling experiments (see Table 2).

Learning rates were tuned on Amazon polarity and BoolQ (using ground-truth labels) to $5 \times 10^{-4}$ for Qwen1.5 0.5B, $2 \times 10^{-4}$ for Qwen1.5 4B, $8 \times 10^{-5}$ for Llama-3 8B, and $4 \times 10^{-5}$ for Llama-3 70B.

We use a fixed batch size of 32, except in our scaling experiments where we approximately mimic the behavior of the OpenAI finetuning API (as of August 2024), which can be seen in Table 2.

While prior work Zhang et al. (2024) suggests that parameter-efficient finetuning does not significantly affect scaling laws for finetuning in multilingual summarization and translation tasks, it is still possible that some of our results could change with full finetuning.

# B  MICROECONOMIC ASSUMPTIONS

We expect that fixed costs will not matter much since they will probably be smaller in magnitude than accumulated marginal costs of labels.

What about scenarios where the cost you invested into training up your labelers or understanding a problem has externalities for other training runs etc?

If classes are extremely imbalanced, you still probably want to train on balanced data, so the marginal cost of an example can just be modeled as the average of the marginal cost of a label from each class, and our results would still apply.

Table 3: Table 1 with $0.50 weak labels.

| Budget | $5 | $17 | $65 | $257 | $1025 | $4097 |
|---|---|---|---|---|---|---|
| Seq SFT | - | 50±2 (0.0) | 56±4 (0.9) | 63±4 (0.9) | 80±2 (0.0) | 87±1 (0.0) |
| +2-shot ICL | - | 51±2 (0.1) | 59±7 (0.9) | 73±10 (0.0) | 83±2 (0.0) | 88±1 (0.0) |
| +log-conf. | - | 50±2 (0.0) | 54±4 (0.9) | 61±3 (0.9) | 81±3 (0.0) | 86±1 (0.0) |
| +unc. sampl. | - | 50±2 (0.0) | 55±4 (0.9) | 62±3 (0.9) | 80±2 (0.0) | 87±1 (0.0) |
| few-shot ICL | 52±4 (1.0) | 58±5 (1.0) | 58±5 (1.0) | 58±5 (1.0) | 58±5 (1.0) | 58±5 (1.0) |

Table 4: Table 1 with $0.01 weak labels.

| Budget | $5 | $17 | $65 | $257 | $1025 | $4097 |
|---|---|---|---|---|---|---|
| +2-shot ICL | 63±7 (1.0) | 75±2 (1.0) | 75±2 (1.0) | 77±3 (0.9) | 84±1 (0.3) | 88±1 (0.0) |
| +log-conf. | 59±2 (1.0) | 69±3 (1.0) | 75±1 (1.0) | 76±3 (0.9) | 82±2 (0.9) | 86±1 (0.0) |
| +unc. sampl. | 60±2 (1.0) | 70±2 (1.0) | 74±2 (1.0) | 79±1 (0.9) | 82±2 (0.9) | 87±1 (0.0) |
| Seq SFT | 60±3 (1.0) | 70±2 (1.0) | 74±1 (1.0) | 77±2 (0.9) | 82±2 (0.3) | 87±1 (0.0) |
| few-shot ICL | 58±5 (1.0) | 58±5 (1.0) | 58±5 (1.0) | 58±5 (1.0) | 58±5 (1.0) | 58±5 (1.0) |

## C    RESULTS

See Tables 3, 1, and 4 for tabular Pareto frontier data at a weak label cost of $0.50, $0.10, and $0.01, respectively. These correspond to the data presented visually in Figure 5.

See figures 6, 7, and 8 for a version of the pareto frontier figure (Figure 5) broken down by dataset.

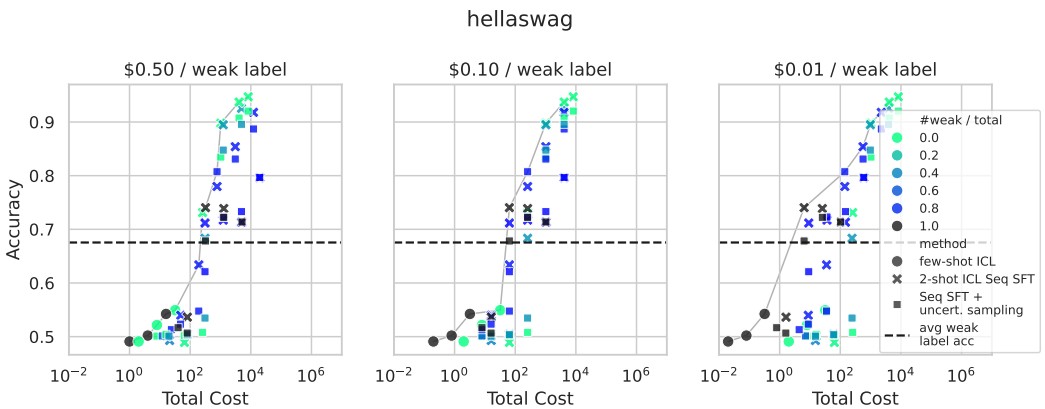

Figure 6: Pareto frontier for Hellaswag, mirroring 5.

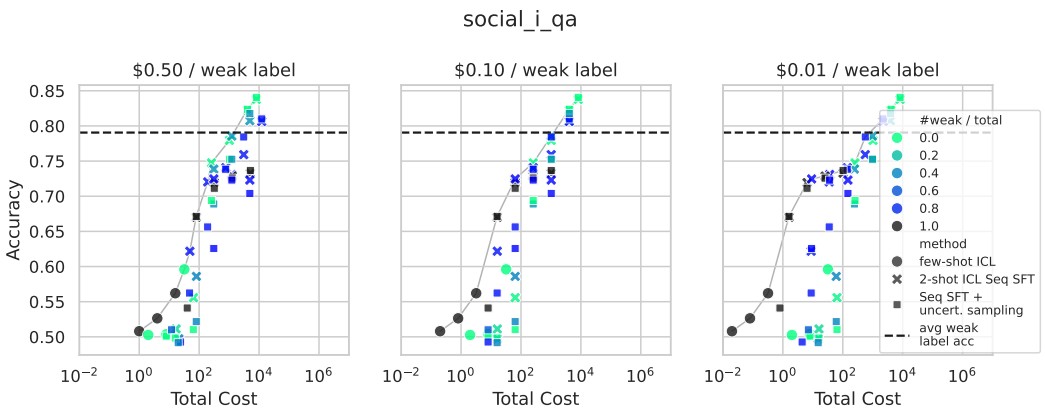

Figure 7: Pareto frontier for SocialIQA, mirroring 5.

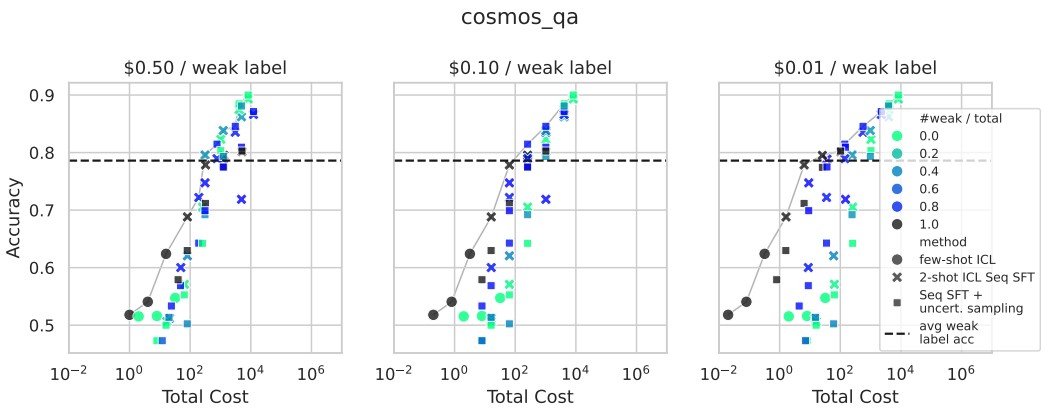

Figure 8: Pareto frontier for CosmosQA, mirroring 5.

