# OpenReview forum: "Balancing Label Quantity and Quality for Scalable Elicitation"
_ICLR.cc/2025/Conference — Submitted to ICLR 2025_

### Official Review · Reviewer_hC7S · 2024-10-17

**Soundness:** 3
**Presentation:** 1
**Contribution:** 2
**Rating:** 3
**Confidence:** 3

**Summary:**

- This paper investigates the quality/quantity tradeoff when finetuning an LLM on cheap low-quality and expensive high-quality labels.
  - The low quality labels are produced by a small LLM that has been finetuned to the task at hand (on a separate datasets), while the high-quality labels appear to be ground-truth labels.
  - Assuming that the cost of low-quality labels is ten times lower than the cost of high-quality labels, the authors find three regimes, depending on the available budget: For small budgets, performance increases in the fraction of low-quality labels used for training. For large budgets, performance instead decreases when more low-quality labels are used. Lastly, for intermediate budgets, performance first increases and then decreases in the percentage of low-quality labels.
- Experiments are done on multiple (binary) NLP classification tasks, and include different training strategies, sometimes combined with few-shot prompting.
- Ablations are conducted using different models, both for label generation and finetuning.

**Strengths:**

- The research question on quality-quantity tradeoffs is interesting.
- While quite specific to the weak-to-strong setting, the experimental setup seems to be well-suited for the research question.
- The experimental results cover multiple tasks and models and are somewhat comprehensive.

**Weaknesses:**

- The paper appears to have been submitted in an unfinished state
  - Different microeconomic assumptions are mentioned as part of the contributions and discussion, but I did not find any discussion of these in the main text (and very little in the appendix).
  - There is a range of minor issues that are unproblematic on their own, but add up:
     - The paper uses the wrong template (ICLR 2024 rather than 2025)
     - The paper containe multiple instances of questions in odd places that appear to be todos by the authors. For example, "All datasets?" at the end of the references, and "What about scenarios where the cost you invested into training up your labelers or understanding a problem has externalities for other training runs etc?" in Appendix B.
     - A bunch of citations are missing details like the arxiv identifier
     - Figures 1 and 2 are presentented in double column without any apparent reason, making things more difficult to read.
   - The logical flow of the writing could be improved at times. For example: "the role of pretraining in weak-to-strong generalization" is mentiond before weak-to-strong generalization is introduced, making the remark difficult to understand.
   - There is also room for improvement in terms of "local" writing clarity. For example:
     - "Results broken down by each of the three datasets can be found in Appendix figures 6, 7, and 8, suggesting that the results hold across tasks and weak label qualities."
        - What result is supposed to "hold" here? I can see that the pareto-fronts have somewhat similar shapes, but what does that imply?
      - The description of the quantity-dominant setting is difficult to parse (perhaps in part due to the dual-column structure).
- Apart from the paragraph on scalable oversight, the related work section is relatively sparse and could benefit from incorporating existing work on sample values [1] and quality/quantity tradeoffs in machine learning [2,3,4].
- While this might be hindsight bias speaking, I do not find the main result, that a mixture of higher quality and lower quality labels is optimal, particularly surprising. For example, while [3] is not fully comparable, in part because high quality labels are selected actively, that work also finds a mixture of higher and lower quality labels to be optimal for training.


[1] Torralba, Antonio, and Alexei A. Efros. "Unbiased look at dataset bias." CVPR 2011. IEEE, 2011.

[2] Sheng, Victor S., Foster Provost, and Panagiotis G. Ipeirotis. "Get another label? improving data quality and data mining using multiple, noisy labelers." Proceedings of the 14th ACM SIGKDD international conference on Knowledge discovery and data mining. 2008.

[3] Chen, D., Yu, Z., and Bowman, S. R. Clean or annotate: How to spend a limited data collection budget. arXiv preprint arXiv:2110.08355, 2021.

[4] Crammer, Koby, Michael Kearns, and Jennifer Wortman. "Learning from data of variable quality." Advances in Neural Information Processing Systems 18 (2005).

**Questions:**

- How and why were the tasks binarized? Does binarization meaningfully affect the results?
- It seems like the gains from using only weak labels are generally quite limited. How does this relate to the results from Burns et al (2023)?
- How do you determine that "256 and 1024 high-quality finetuning examples do not reliably elicit knowledge from Llama-3-8B, but elicit most of Llama-3-70B’s knowledge."?
- "The quality of in-context examples is inconsequential, while the quality of SFT examples matters substantially."
  - Is this refering to a specific regime?  The top/bottom left of figure 4 seems to show a big difference, at least for 2/8 in-context and few SFT samples.
- Is there a typo in the caption of Figure 3? The accuracy of the weak labels displayed in the figure appears to be larger than 70.2%. Or is this the difference between training/test accuracy?
- "We use 3 random seeds, except for training runs where the smaller stage takes less than or equal to 10 examples, in which case we use 7 random seeds"
  - How do you get less than 10 examples when the minimum budget is at 16?
- Regarding the GPT-4 results, how well does GPT-4 perform on the tasks in a zero-shot manner?

---

### Official Review · Reviewer_fvP4 · 2024-10-22

**Soundness:** 3
**Presentation:** 2
**Contribution:** 2
**Rating:** 5
**Confidence:** 3

**Summary:**

The paper explored different ellicitation techniques on improving an NLP binary classification tasks under some fixed budgets. Under this budget, one can either have a lot of low quality data, or a few high quality data, or a mixed between the two. Using these tasks, the author explored the pareoto frontier between mixing the low and high quality data, finding that no pure strategy dominates.

**Strengths:**

The question on the trade-off between quantity and quality is interesting; and the finding that no pure strategy (few high quality data and much low-quality data) is clear. The results are supported by extensive experimentation.

**Weaknesses:**

My biggest concernis that the insights of the paper is not very clear. As it is true that there is a trade-off between quantity and quality, it is unclear how the specific finding in this paper can be generalizable to tasks beyond NLP binary classifications showcased in the paper. In this case, it might make sense to come up with a scaling law that captures this tradeoff.

**Questions:**

Why choose NLP classification task as the one to test this tradeoff? Will the result holds for other tasks? If this is unclear, what should the main takeaway for the reader of the work to be?

---

### Official Review · Reviewer_acFA · 2024-11-03

**Soundness:** 2
**Presentation:** 2
**Contribution:** 2
**Rating:** 3
**Confidence:** 3

**Summary:**

This paper considers the task (termed "scalable oversight") of training LLMs on problems that human judgment is unreliable or expensive. The paper conducts experiments that vary {the type of large models, the datasets and associated tasks, the training methods}. Using results from these experiments, the paper observes different regimes where it can be more effective to only collect high-quality data, low-quality data, or a mix of both.

**Strengths:**

1. The paper poses an important and interesting question about what the best methodology (i.e., what type of data to collect and what training method to use) is in this challenging regime of "hard" tasks where human data is unreliable or expensive. Improving LLMs' ability to generalize to these tasks has lots of potential in making positive societal impacts.

2. The paper presents a wide range of experimental settings and results, which provide useful information for future practitioners working on improving LLMs on these hard tasks.

**Weaknesses:**

While I appreciate the efforts and empirical results by the paper, I feel that this paper looks more like an engineering technical results for the following reasons:

1. Scientific rigor:

(a) Please include error bars on all reported plots. If such errors are not available, please provide discussions on the statistical significance of the reported observations.

(b) The experimental observations can be made more precise. For example, while the paper claims that "$256-$512" is the mixed regime in Figure 1, this appears very task dependent in Figure 2.

(c) The discussion on limitations can be more in depth. For example, a distinction can be made on low/high-quality data provided by human vs. smaller LMs (as in the experiments). Does the paper specifically study training data provided by LMs? Does this generalize to low/high-quality data provided by people or not? Also in the experiments, the smaller LMs are trained on other data ("we generate weak labels using small LMs that have been finetuned on the task"), which confounds the conclusions made in the paper about low/high-quality data in the experiments.

2. Unsubstantiated contributions:

(a) The second claimed contribution includes "microeconomic assumptions", but as far as I can tell, this is not investigated in the paper. Appendix B appears unfinished.

(b) I also think the third claimed contribution is very limited. While research framing can be a valid contribution, in this case, the problem of data usage and training methods is a well-known research topic and extensively studied in the past. The Pareto frontier is essentially a piece of empirical result that provides some practical evaluation and guidance on what training methods work better in what settings.

2. Lack of context:

I find the background knowledge introduced in this paper insufficient for audience beyond LLM practitioners. For example, what are the tasks "BoolQ, HellaSwag, SciQ, Cola, ..."? What's "LoRA" for supervised finetuning? What does it mean to "early-stop based on validation AUROC"? Why are "Qwen 1.5-0.5B and Qwen1.5-4B" chosen for the initial set of experiments? What's a new "head" for training? What's the procedure for "few-shot prompting" and what are "in-context examples"?

3. Insufficient literature review:

The literature review only provides citations on very recent advances specifically tied to LLM development. I personally find it insufficient without giving credits or contextualizing the research question with respect to other lines and domains of work. Two of such domains are:

(a) The field of crowdsourcing primarily concerns how to make use of noisy human data, which is closely relevant to the paper.

(b) The field of economics computation literature concerns data valuation and pricing, which is also closely relevant to the paper.

4. Clarity (minor):

(a) In the result section, the paper mentions "quality of weak labels". What does this mean? I thought weak labels mean low quality in this paper.

(b) I suggest the paper using different terms than "quantity-dominant" and "mixed". These terms are confusing because the quantity-dominant regime cannot afford enough weak labels, and the mixed regime can afford enough weak labels.

(c) The wording "scaling results" in Section 4.1 is vague. What results does this refer to?

**Questions:**

1. Could the paper provide more explanations about how the "true" labels for evaluation is obtained? If people cannot provide accurate high-quality labels, how do we evaluate machine performance on these hard tasks? Is there a chicken-and-egg problem here?

2. Could the authors comment on how the reported results compare with human performance?

3. What's the definition for these "hard" tasks? One precise piece of information mentioned in the paper is that people are "less than 90% accurate", but an accuracy of 90% does not appear too hard to me.

4. Addressing my other main concerns in the "Weaknesses" section would be helpful.

---

### Meta-Review · Area_Chair_R6oW · 2024-12-19

**Metareview:**

This paper considers the task of training LLMs to solve problems where human judgment is expensive and there is a need to balance the quantity and quality of labelled examples. The main strengths of the paper mentioned in the reviews is (1) its setting, which is motivated by many real world applications, and (2) the extensive experiments reviewing a wide variety of tasks.

The main weaknesses raised relate to the state of the paper. Generally, as mentioned by hC7S, the paper seems to have been submitted in an unfinished state. Some concrete issues mentioned: The comparison w.r.t existing works require more work (suggestions are given by both acFA, hC7S), the experiments should be better presented (see comments by acFA). Given this, I recommend rejecting the paper since in its current form since it isn't ready to be published.

**Additional Comments On Reviewer Discussion:**

n/a

---

### Decision · Program_Chairs · 2025-01-22

Reject